# Automatic segmentation of stroke lesions in non-contrast computed tomography with convolutional neural networks

**Anup Tuladhar**[*][1]                                                    ANUP.TULADHAR@UCALGARY.CA

**Serena Schimert**[*][1]                                               SERENA.SCHIMERT@UCALGARY.CA

**Deepthi Rajashekar**[1]                                           DEEPTHI.RAJASHEKA1@UCALGARY.CA

[1] *Department of Radiology, Cumming School of Medicine, University of Calgary, Canada*

**Helge C. Kniep**[2]                                                               H.KNIEP@UKE.DE

**Jens Fiehler**[2]                                                                  FIEHLER@UKE.DE

[2] *Department of Diagnostic and Interventional Neuroradiology, University Medical Center Hamburg-Eppendorf, Germany*

**Nils D. Forkert**[1]                                                    NILS.FORKERT@UCALGARY.CA

## Abstract

Manual lesion segmentation for non-contrast computed tomography (NCCT), a common modality for volumetric follow-up assessment of ischemic strokes, is time-consuming and subject to high inter-observer variability. Our approach uses a combination of a 3D convolutional neural network (CNN) combined with post-processing methods. A total of 291 multi-center clinical NCCT datasets were used: 204 for CNN training, 48 for validation and developing post-processing methods, and 39 for testing. The testing datasets were from centers that did not contribute to the training and validation sets, and were segmented by two or three neuroradiologists. We achieved a mean Dice score of 0.42 on the out-of-distribution test set, which was significantly improved to 0.45 with post-processing. The automatically segmented lesion volumes were not significantly different from the lesion volumes determined by manual segmentations from multiple observers. As the model was trained on datasets from multiple centers, it is broadly applicable and publicly available.

**Keywords:** stroke, computed tomography, segmentation, deep learning, CNN

## 1. Introduction

NCCT is the most common imaging modality for volumetric assessment of stroke lesions (Eswaradass et al., 2016). Manual lesion segmentation in NCCT images is time consuming and associated with high inter-observer variability. Semi-automatic lesion segmentation tools have been developed (Kuang et al., 2019), but still require human interaction while previous work on automatic NCCT lesion segmentation is very limited (Sales Barros et al., 2019; Fuchigami et al., 2020).

CNNs show superior performance for various segmentation tasks in medical imaging because of their ability to learn complex patterns and relationships in the data (Zaharchuk et al., 2018). The use of multi-scale features and three-dimensional kernels (Kamnitsas et al., 2017) would allow an automated segmentation algorithm to take advantage of the

---

[*] Contributed equally

spatial contiguity of stroke lesions while maintaining localized focus. However, for stroke lesion segmentation, these methods have only been applied to magnetic resonance imaging (MRI) (Chen et al., 2017; Wu et al., 2019; Liu et al., 2019) or computed tomography perfusion and angiography datasets (Öman et al., 2019; Kasasbeh et al., 2019). Multi-scale 3D CNNs for lesion segmentation have not been evaluated in NCCT datasets, despite its common application in follow-up stroke imaging.

The aim of this work was to train and evaluate a CNN model for stroke lesion segmentation in NCCT datasets. To improve upon CNN segmentations we investigated post processing methods. We tested the models generalizability by evaluating it on an out-of-distribution holdout test set acquired from entirely different studies from those used in training and validation (Figure 1A).

## 2. Materials and Methods

A total of 291 clinical follow-up NCCT datasets acquired at 29 centers and corresponding manual segmentations were available. The in-slice resolution ranged from 0.355 to 0.637 mm, the slice thickness ranged from 1.00 to 10.0 mm, while the number of slices ranged from 10 to 141. 204 datasets were used for training of the 3D CNN-based lesion segmentation model, 48 datasets for validation, and 39 datasets for testing. The 39 out-of-distribution holdout datasets used for testing were acquired at seven centers not contributing to the training or validation sets and were manually segmented by two (19 datasets) or three (20 datasets) neuroradiologists.

All datasets were preprocessed identically to ensure data consistency across the different scanners and acquisition protocol. A 3D multi-scale CNN was trained with 204 datasets and corresponding lesion segmentations using the previously described DeepMedic (Kamnitsas et al., 2017) framework (v0.7.3). The CNN-based lesion segmentations were post-processed to improve accuracy. In brief, post-processing consisted of a connected component analysis to exclude small lesion components, most likely caused by noise artifacts, and an automatic hole-filling approach. The minimum lesion size and hole-filling kernel size were systematically optimized using the validation datasets, resulting in final values of 1.5 mL and 3 voxels, respectively. Automatic segmentations were evaluated using the Dice similarity coefficient (DSC) and lesion volume. Intra-class correlation coefficients (ICC) were used to assess inter-rater reliability in lesion volume estimates (Shrout and Fleiss, 1979). Further details on data pre-processing, model design, training parameters, post-processing and model evaluation are available in Appendix A.

Results are reported as mean $\pm$ standard deviation (SD) or median [interquartile range] as appropriate. The Friedman test with Dunn's multiple comparison post-hoc correction was used for comparisons. Statistical significance was set as $P < 0.05$. All statistical analysis was performed with Graphpad Prism 8.4.

## 3. Results and Discussion

The median lesion volumes for the training and validation sets were 40.4 [14.1–96.3] mL and 41.5 [20.0–107.1] mL, respectively. As the out-of-distribution holdout test set was segmented by multiple observers, the manual segmentation lesion volume for each example

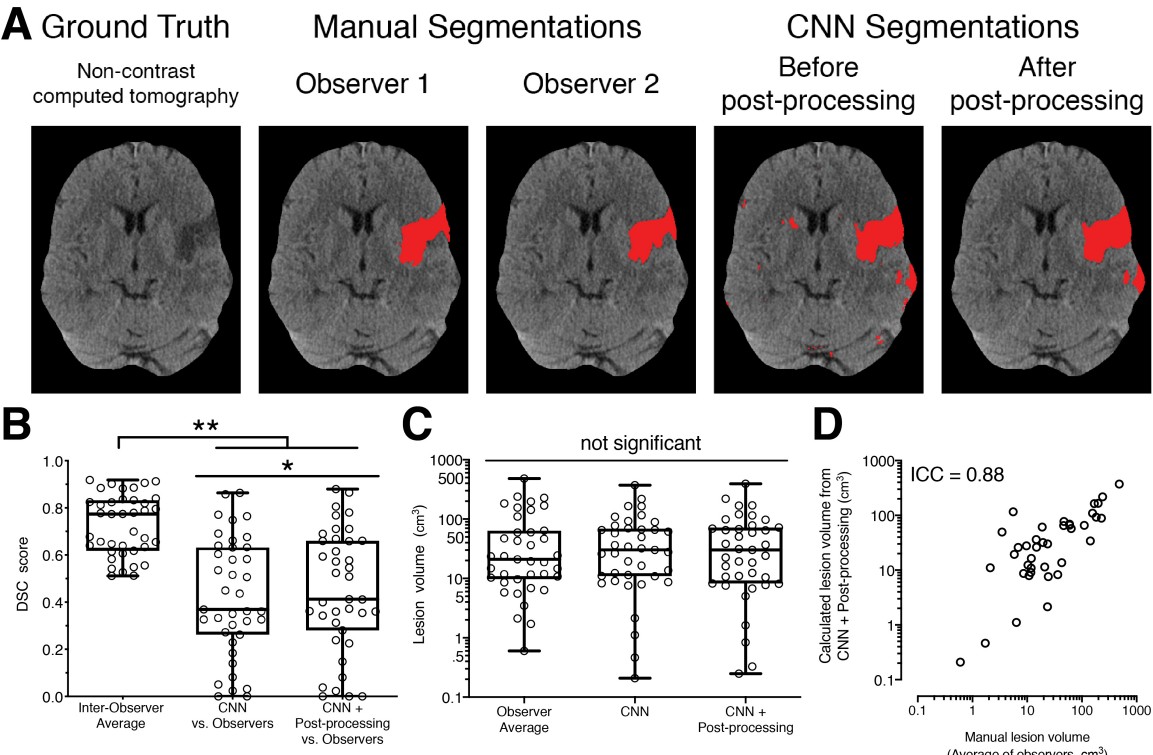

Figure 1: Comparison of CNN-based segmentations on an independent holdout test set segmented by 2 independent observers. **A**, Example of segmentations. **B**, DSC score. **C**, Calculated lesion volumes. **D**, Agreement of lesion volumes estimates between human observers and CNN-based segmentations after post-processing.

was defined as the average volume calculated across observers. The median lesion volume for the test set was 20.9 [9.7–63.7] mL, which is considerably lower compered to the training and validation sets.

The CNN achieved a mean DSC of 0.42 ± 0.25 compared to manual segmentations (Figure 1B). Post-processing of CNN-based segmentations significantly improved the DSC to 0.45 ± 0.26 ($P < 0.05$). Both were lower than the inter-observer DSC of 0.73 ± 0.13 ($P < 0.01$). No significant difference was found between the lesion volume estimates of manual segmentations (20.9 [9.7–63.7] mL) and and the CNN-based lesion segmentation before (30.1 [10.9–68.9] mL) and after post-processing (30.2 [8.2–72.2] mL) ($P > 0.05$) (Figure 1C). Bland-Altman analysis showed minimal systematic bias of the lesion volume estimates by automatic segmentations compared to manual segmentations (Additional Figure 2). Importantly, lesion volume estimates from CNN segmentations showed excellent agreement with manual segmentations. The ICC for CNN lesion segmentations was 0.86, which was further improved by post processing to 0.88 (Figure 1D). The agreement between observers for manual segmentations was lower, with an ICC of 0.80.

As NCCT is a standard imaging procedure available in most stroke centers for follow-up assessment, an automatic lesion segmentation pipeline for this modality is of high demand. CNN models for follow-up lesion segmentation have primarily been investigated for MRI (Kamnitsas et al., 2017; Chen et al., 2017; Wu et al., 2019; Liu et al., 2019), achieving higher DSCs (0.67–0.79) than seen in our study. Nevertheless, the proposed method is very promising given that lesion segmentation in NCCT is more challenging compared to typical MRI follow-up sequences such as diffusion weighted imaging (Fiebach et al., 2002) as the ischemic changes are subtler.

This study demonstrated the successful use of a CNN-based lesion segmentation in clinical NCCT datasets. Though the voxel-wise agreement of the CNN-based segmentations was inferior to the inter-observer agreement, the corresponding lesion volumes were not different from manual segmentations and was in excellent agreement with them. This suggests a potential application of the model for volumetric assessment of follow-up lesions. Within this context, it is reassuring that the agreement between automatically and manually segmented lesion volumes was higher than the inter-observer agreement between manual segmentations. Providing consistent results is an advantage of automatic segmentation algorithms, thereby reducing variability between sites or studies. This lays the foundation for developing automatic lesion analysis tools for NCCT images and can contribute toward consistent and high-throughput analysis of large multi-center studies. The full journal article associated with this work can be found in (Tuladhar et al., 2020a). The trained model is freely available for NCCT lesion segmentation from http://dx.doi.org/10.21227/jps9-0b57 (Tuladhar et al., 2020b).

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

## Acknowledgments

We would like to thank Michael D. Hill, Mayank Goyal and Andrew M. Demchuk for sharing data from the ESCAPE trial (NCT01778335) that formed a portion of training and validation sets used in this study.

This work is supported by the Heart and Stroke Foundation of Canada Grant in aid [G-17-0018368], the T. Chen Fong Fellowship in Medical Imaging Science (to AT), the Natural Sciences and Engineering Reseasrch Council of Canada (NSERC) Postdoctoral Fellowship (to AT), and the University of Calgary Program for Undergraduate Research Experience (PURE) award (to SS).

## Appendix A. Supplemental Methods

### A.1. Segmentation evaluation

The Dice similarity coefficient (DSC) was used as the primary outcome measurement for evaluation of the automatic lesion segmentations. The DSC measures the overlap between two segmentations and is defined between 0 and 1, whereas 1 indicates perfect consensus. The DSC is calculated as $DSC = (2 * |A \cap B|)/(|A| + |B|)$, where A and B are segmentations from manual and automated segmentations, respectively.

For inter-observer DSC, A and B were the segmentations from observer A and observer B, respectively.

DSC scores for training data were obtained by 10-fold cross-validation. Samples were randomly assigned to test folds. The training samples were evaluated when they were a part of a fold's test data.

DSC scores and lesion volumes from automatic segmentations, for the validation and holdout test set were obtained using a single CNN model that was trained on the entire training data.

### A.2. NCCT scan pre-processing

As the NCCT images were acquired from multiple centers, with differing scanners and imaging protocols, the datasets had to be pre-processed to ensure consistency.

First, the bone structures were removed from each dataset, retaining only the brain tissue in the images. To remove the bone structures, which have high Hounsfield values, a six-step procedure following the approach described by Muschelli et al (Muschelli et al., 2015) was performed in a slice-wise manner. The approach was implemented using the Insight Segmentation and Registration Toolkit (ITK) (Yoo et al., 2002). A Gaussian filter with a variance of 4 pixels was used to smooth each slice. Next, the intensities were thresholded between 0 and 100 Hounsfield units and a circular structural element with a radius of 1 pixel was used to erode the resulting segmentation. Afterwards, the largest connected component in each slice is extracted and a circular structural element with a radius of 1 pixel was used to dilate this component in order to create a brain mask for the slice. Finally, after performing the first three steps in each slice, the masks from each slice are combined into a final mask for the entire image and any holes in this final mask are filled using the VotingBinaryHoleFillingImageFilter in ITK.

Second, the images were thresholded between 0 and 100 Hounsfield units to remove noise and hypo- or hyper density artifacts. Finally, the images were normalized to unit variance to account for potential differences in scanner tube potential and different reconstruction algorithms. All images in the training, validation, and test datasets underwent the same pre-processing procedure.

### A.3. 3D Convolutional Neural Network Architecture

The CNN used in this work is based on the DeepMedic model proposed by Kamnitsas et al. (Kamnitsas et al., 2017) and modified for NCCT stroke lesion segmentation. The network parameters were optimized with cross validation. We used a total of 11 layers. The first eight layers consist of three parallel convolutional pathways for processing the images at multiple scales. The multi scale pathways were created by using down sampled versions of the NCCT images (by factors of 3x and 5x) as inputs to the parallel convolutional pathways, in addition to the original image. Each parallel pathway has eight convolutional layers consisting of 30, 30, 40, 40, 40, 40, 50, and 50 feature maps and uses convolutional kernels of size 3x3x3. Additionally, residual skip connections between layers two and four, between layers four and six, and between layers six and eight are used in each parallel pathway. The ninth layer combines the three multi scale pathways together by using the concatenated outputs from layer eight of each parallel pathway. Layer nine uses 3x3x3 convolutional kernels and has 250 feature maps. Layer ten is a fully connected convolutional layer with 1x1x1 convolutional kernels and 250 feature maps. Additionally, a residual skip connection between layers eight and ten was used. The final softmax classification layer, layer eleven, produces the lesion probability maps. A threshold of $> 0.5$ is used to binarize the probability map to a final lesion segmentation.

### A.4. 3D Convolutional Neural Network Training

All CNN model training was performed in Python 2.7 on Compute Canada and Calcul Quebec computing clusters. The DeepMedic framework (v.0.7.3), was used for model training. The DeepMedic framework performs model training on image segments extracted from the original image, rather than the entire image. In this work, segments of 37x37x37 were used. The network was trained for 35 epochs with a batch size of 10. Each epoch was divided into 20 sub epochs, within which 1000 image segments were extracted and used for model training. An initial learning rate of 0.001, which decreases through training using a polynomial decay function, was employed. Root mean square propagation was used as the optimizer. L1 and L2 regularizations of 0.000001 and 0.0001 were used, respectively. Data augmentation consisted of mirroring along the sagittal axis.

### A.5. Post-processing

Post-processing was done exclusively using the ITK toolkit via the SimpleITK implementation. Post-processing was conducted on the binary lesion segmentations produced by the trained DeepMedic model. These segmentations were passed through the CastImageFilter with OutputPixelType set at 4 (32-bit signed integer) in order for the images to be compatible with the necessary filters.

Using a connected components analysis, components below 1.5 mL in volume were removed. The exception was in segmentations where the largest connected component was smaller than the cutoff, in which case no cutoff was used. Finally, hole-filling with a radius of 3 voxels was used to fill gaps within the segmentation.

The validation dataset was used to optimize the minimum object size threshold and the hole-filling kernel radius. The minimum object size threshold was optimized first, by varying the threshold range from 0.3 mL to 2.5 mL. The value that maximized the DSC

was 1.5 mL. Using this threshold, the hole-filling radius was optimized next using values of 2, 3, 5, 7, and 10 voxels. As hole-filling causes the segmented lesion volumes to grow, and subsequently increased the error in lesion volume estimates, both the DSC and lesion volume error were considered when choosing the optimal value. More precisely, the DSC was maximized while the lesion volume error was minimized. The optimal radius was found to be 3 voxels.

## Appendix B. Additional Figures

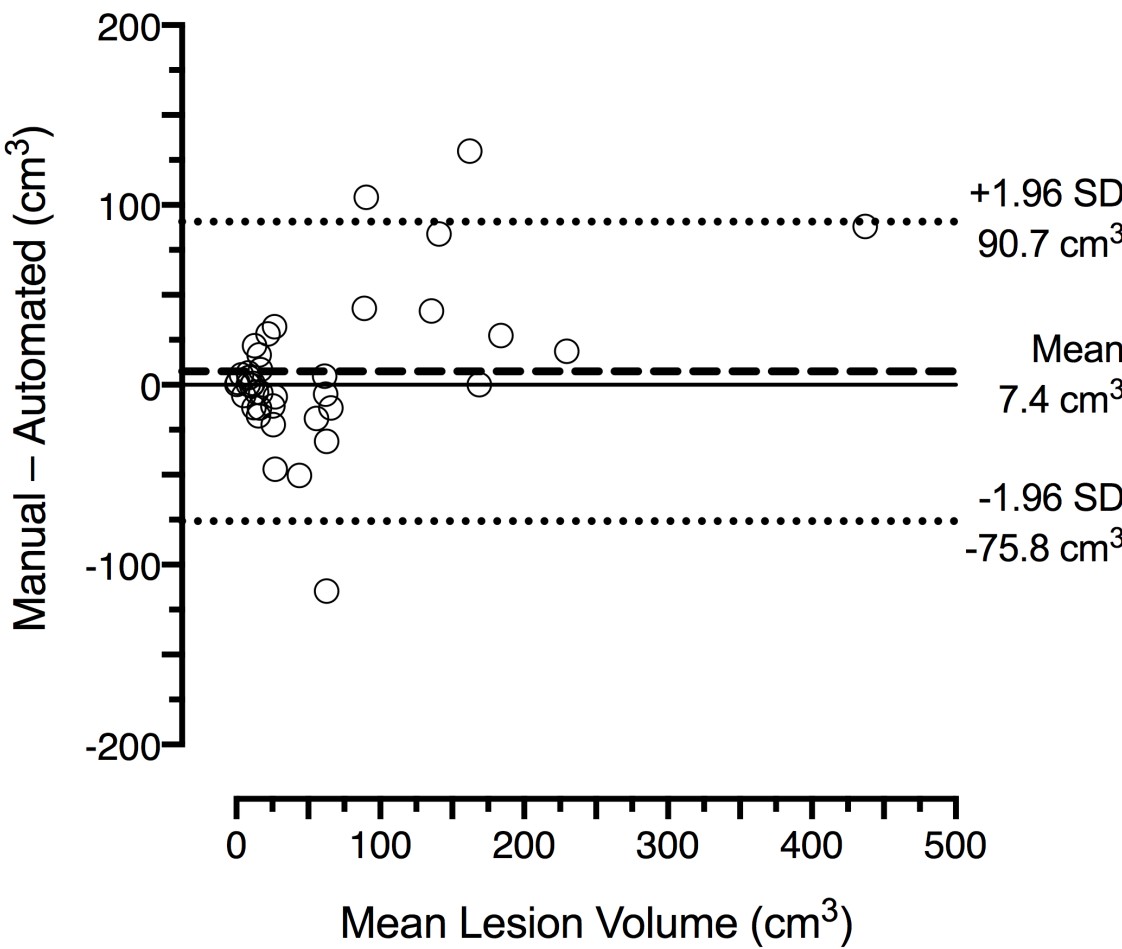

Figure 2: Bland-Altman agreement analysis of test set lesion volumes from manual segmentations and CNN segmentations with post processing, showing minimal bias. SD: standard deviation.

