# OpenReview forum: "Automatic segmentation of stroke lesions in non-contrast computed tomography with convolutional neural networks"
_MIDL.io/2020/Conference — MIDL 2020_

### Official Review · AnonReviewer1 · 2020-03-06
**NCCT lesion segmentation using DeepMedic**

**Rating:** 3
**Confidence:** 4

**Review:**

Authors present their work on applying DeepMedic to segment lesions on NCCT images. The topic of the work is relevant and interesting. The evaluation on two data sets from hospitals not involved in training is a major strength of this work. The major weakness is the limited methodological novelty, since this is merely a validation study.

As I said, the evaluation is very strong, because authors used images from two hospitals not involved in training. This gives confidence that the performance of this method is reproducible in other studies. This is a major strength and unfortunately not very common in this field.

The ratio of train/val/test data is quite skewed towards train/val: 204/48/20. Do you really need 204+48 images to train DeepMedic to achieve this performance? It might be very interesting to see whether the performance on the 20 test images changes when using less training data. If this method can be trained with less data, it would make it even more attractive to use.

20 test images, although from different hospitals, is still quite limited. Personally, I would have opted to include much more test data and use less training data. Perhaps in future work authors can extend the test set and demonstrate performance on a larger data set.

It might be interesting to specifically look at small lesions? It is usually not very hard to detect / segment large lesions and Dice is always high for larger lesions. Automatic solutions might be key in finding small lesions, since these are also hard to spot visually by an observer. Can authors comment on the performance of small lesion (e.g. below median size)?

Please report the inter-rater dice also in the text, I could only find it in the figure.

---

### Official Review · AnonReviewer2 · 2020-03-09
**A combination of CNN and post-processing methods for lesion segmentation for non-contrast computed tomography images**

**Rating:** 3
**Confidence:** 4

**Review:**

Pros:
* The segmentation output from the DeepMedic framework was corrected with post-processing, such as connected component analysis and hole-filling.

* The median dice of the reported method show improved performance over two manual graders. The correlation of lesion volumes between the human graders and the CNN-based segmentation was statistically non-significant.

*Further, the authors have performed extensive statistical analysis to justify the significance of the method.

Minor comments:
* Do NCCT datasets contain 272 samples (not dataset)? Are 204 samples (not datasets) used for training?

* In section 3, No significant difference ……… "were found" is repeated twice.

---

### Official Review · AnonReviewer3 · 2020-03-12
**Short application paper with solid statistics**

**Rating:** 3
**Confidence:** 4

**Review:**

The presented paper describes the application of the pre-existing "DeepMedic" codebase for segmenting ischemic stroke lesions in non-contrast CT images. While the authors selected "both" as paper type, I do not see any methodological contributions. There is some standard pre-and post-processing; the preprocessing is only described in the appendix and mostly consists of skull stripping and HU range selection, the post processing is described in more detail but only consists of hole filling and removal of tiny isolated components.
On the positive side, I like to point out that the authors did not just present average dice coefficients, but performed statistical tests and decided to describe the performance based on quantiles. Furthermore, they had two observers annotate the dataset, and the dataset comes from 24 sites and shows quite some variability in imaging parameters. So, I believe the final evaluation results to be fairly realistic, much more so than with the average paper.
Critically speaking, the post processing applied includes a fully-connected CRF; at least that's included in DeepMedic to the best of my knowledge. It's interesting that the CRF cannot learn itself that isolated components of up to 3 voxels should be removed. Speaking of which, I found it strange that the unit voxels is used for removing cruft, but the hole filling threshold is given in ml. When the main paper merely stated that the "datasets were preprocessed identically to ensure data consistency", I thought that referred to the voxel size, since that varied considerably in the dataset. (Yet, no resampling was applied.) It might have been better to at least hint at the kind of preprocessing / harmonisation. I think masking the CNN's input both during training & inference would've made sense, but it is not mentioned.
The conclusion that "[the strong correlation] suggests a potential application of the model for volumetric assessment of follow-up lesions" is a bit far-fetched; for instance, a lesion annotated to be around 6ml by human experts is segmented with >100ml, and a lesion with 40ml is underestimated to be only 10ml.
Overall, I think it is a nice application paper. It does not present any technical novelty, but the evaluation is sound. (The authors also announce to release the model together with the final manuscript. Only the dataset is obviously private.)

There is a duplicate "were found" in the last sentence of the first paragraph in section 3.

---

### Official Review · AnonReviewer4 · 2020-03-12
**interesting perspective however novelty is somewhat limited**

**Rating:** 2
**Confidence:** 4

**Review:**

The paper demonstrates a CNN-based segmentation network for non-contrast CT images. The performance of the network is evaluated with/without post-processing and compared to expert annotators. The results indicate that the proposed method produces results positively correlated with the experts and the segmentation accuracy, measured as Dice score, can be improved by post-processing.

The authors point out that an interesting point that NCCT images are common in stroke imaging while segmentation methods are mainly proposed for MRI images. However, it would be more helpful if the authors could give some advantages of NCCT in clinical use, in terms of cost, accessibility etc, and discuss why methods are more often evaluated on MRI images, could it be that NCCT images are more difficult to acquire, or public datasets are scarce to promote more research, or annotation on NCCT is less accurate due to its subtle contrast change?

The paper reads more like an evaluation of existing method, as the segmentation method is mostly built on DeepMedic. The correlation analysis to the neurologists' annotations is very interesting. It would be more interesting if more segmentation methods can be evaluated in this way and compared with one another on NCCT images and present the results as a baseline to promote further research.

---

### Meta-Review · Area_Chair1 · 2020-04-05
**MetaReview of Paper161 by AreaChair1**

**Rating:** 3

**Metareview:**

This paper shows promising results when applying developed architecture to imaging types usually not deemed sufficiently informative. Despite the limited testing and validation pool, the statistics appear well performed.

**Paper Type:**

validation/application paper

---

### Decision · Program_Chairs · 2020-04-11

Accept